# Geochemical and Isotopic Compositions and Geothermometry of Thermal Waters in the Magumsan Area, South Korea

**Chan-Ho Jeong [1], Byeong-Dae Lee [2,\*], Jae-Ha Yang [3,\*], Keisuke Nagao [4], Kyu-Han Kim [5], Sang-Won Ahn [1], Yong-Cheon Lee [1], Yu-Jin Lee [1] and Hyeon-Woo Jang [1]**

1   Department of Construction and Disaster Prevention Engineering, Daejeon University, Daejeon 34520, Korea
2   Groundwater Research Center, Korea Institute of Geoscience and Mineral Resources, Daejeon 34132, Korea
3   EGI Consulting Co., Incheon 22698, Korea
4   Division of Polar Earth System Science, Korea Polar Research Institute, Incheon 21990, Korea
5   Department of Science Education, Ewha Womans University, Seoul 100-744, Korea
\*   Correspondence: blee@kigam.re.kr (B.-D.L.); egiconsl@gmail.com (J.-H.Y.);
     Tel.: +82-42-868-3088 (B.-D.L.); +82-10-6408-1872 (J.-H.Y.)

**Abstract:** The Magumsan thermal waters of the southeastern Korean Peninsula are pumped out of six deep wells (average depth, 300 m) at temperatures of 30.8–49 °C. The thermal waters are chemically classified into two groups: $NaHCO_3$ type (<31 °C) and NaCl ($HCO_3$, $SO_4$) type (>40 °C), both of which have chemical compositions that are distinct from local groundwater (Ca–$HCO_3$ type). $\delta^{18}O$ and $\delta D$ values suggest that the thermal waters originate from meteoric water and they are isotopically fractionated by silicate hydration or $H_2S$ exchange. $\delta^{34}S$ values (+7.0 to +15‰) of dissolved sulfate in the thermal waters reflect enrichment in $^{34}S$ through kinetically controlled oxidation of magmatic pyrite in the thermal aquifer and mixing with paleo-seawater. On the $^3He/^4He$ vs. $^4He/^{20}Ne$ diagram, the thermal waters plot along a single air mixing line of dominant crustal He, which indicates that the heat source for the thermal waters is non-volcanogenic thermal energy that is generated from the decay of radioactive elements in crustal rocks. Chalcedony geothermometry and thermodynamic equilibrium calculations using the PHREEQC program indicate a reservoir temperature for the immature thermal waters of 54–86 °C and 55–83 °C, respectively.

**Keywords:** thermal waters; chemical composition; reservoir temperature; $\delta^{34}S$; meteoric water; $^3He/^4He$ ratio; geothermometers; thermodynamic equilibrium

## 1. Introduction

Hot-spring thermal water has been developed at about 400 sites in South Korea for spa and medical purposes, with −68% of the springs presenting temperatures of 25–30 °C.

Thermal water in the Magumsan area was first developed as a commercial hot spa in 1926 from a deep well in andesitic rocks of the Cretaceous Kyeongsang basin. Nowadays, the thermal waters in this area come from six deep wells (average depth −300 m), which are distributed in an area of 200 m diameter. The Magumsan thermal water is reported to have a chemical composition, including $Na^+$, $Cl^-$, and $SO_4^{2-}$ as the major ions, with a maximum temperature of 55 °C [1]. Although Kim and Nakai analyzed the $\delta^{34}S$ of dissolved sulfate in one sample of the Magumsan thermal waters [2], they suggested that sulfate S originated from mixed sources of seawater and magmatic pyrite sulfate. Park [3] explained that the chemical composition of the Magumsan thermal water with high Cl has been controlled by the flushing effect of saline thermal water by fresh water, owing to the seawater regression.

However, until the present day, helium and sulfur isotopic, and geothermometric methods have not been applied in elucidating the sulfur origin, heat source, geothermal reservoir temperature, or mixing ratio of thermal and cold water.

Geothermal systems in non-volcanic stable continental regions are unlikely to have magmatic heat sources. Rather, their heat source is mostly of crustal origin and is associated with the decay of radioactive elements (e.g., $^{238}$U, $^{232}$Th, $^{40}$K) [4]. The circulation depth of meteoric water and the presence of cap rock preventing heat loss from the system controls maximum reservoir temperatures of such geothermal systems. Another heat source is that developed along interfaces between lithospheric sectors, where friction and viscous shearing forces along boundaries are converted into heat [5].

Helium isotopic compositions of thermal water can provide critical constraints for heat sources in geothermal systems, owing to (1) the distinctive He isotopic ratios in mantle and crustal reservoirs, (2) contemporaneous release of heat and He from a magmatic mass or from decay of radioactive elements in the crust, and (3) the upward transport of mantle and crustal He accompanying transport of heat within the crust [4]. Jeong et al. [6–8] and Park et al. [9] have conducted helium isotopic applications for thermal water in South Korea to reveal heat sources by the determination of mixing ratios among three end-members (i.e., atmosphere, crust, and mantle) via $^{3}$He/$^{4}$He–$^{4}$He/$^{20}$Ne ratio plots. They suggested that the heat source of thermal alkaline waters in South Korea is mainly constrained by an energy source that is produced from decay of radioactive elements in granitic rocks, which is based on dominant contribution of crustal helium ($^{4}$He) with different ratios of $^{3}$He contribution from the deep source, such as magma or mantle according to thermal sites.

Cation and silica geothermometers, the Na–K–Mg ternary diagram, the silica–enthalpy model, and the thermodynamic equilibrium method have widely been used as tools in estimating the reservoir temperatures of geothermal systems, based on chemical analyses of the thermal waters [10–13]. After a cross validation of the results from each methods in this research, suitable methods were adapted to estimate the reservoir temperature of Magumsan thermal waters, because of the different range of their applicable temperature and given chemical data condition.

The aims of the present study were to determine the chemical composition, recharge origin, and sulfur origin (as sulfate) of the Magumsan thermal waters, in order to quantitatively estimate the ratios of He of different origins in thermal waters, including crust, mantle, and atmosphere, and to identify the major subsurface heat source of the geothermal reservoir. The geothermal reservoir temperature and mixing ratio of thermal water and shallow cold water were estimated while using geothermometers, the silica–enthalpy model, and the thermodynamic equilibrium method, which have their intrinsic applicable temperature range.

## 2. Study Area

The Korean Peninsula is situated on the southeastern margin of the Eurasian Plate, in a transitional setting between island-arc and back-arc terranes that formed through the subduction of the Philippine Sea Plate beneath the Eurasian Plate (Figure 1a). The geology comprises five main geological provinces: the Gyeonggi and Youngnam massifs, the Ogcheon Belt, Jurassic–Cretaceous granitoids, and the Kyungsang basin (Figure 1b). Precambrian basement rocks, which are exposed in the Gyeonggi and Youngnam massifs, comprise 2.7–1.1 Ga high-grade gneisses and schists. These massifs are separated by the Ogcheon Belt, which is a northeast-trending fold-and-thrust belt that comprises low- to medium-grade metasedimentary rocks [14]. Most of the thermal waters in South Korea are related to the fault zone in the Jurassic and Cretaceous granitoids, whereas thermal water in the Gyeongsang basin, which mainly comprises andesitic and sedimentary rocks, has been locally developed in areas with much higher than average surface heat flow of 66.6 mW m$^{-2}$ [15]. The Magumsan area, in the southeastern Korean Peninsula, is surrounded to the east by Mt. Cheonma (365 m) and to the west by Mt. Magum (276 m). The average annual temperatures recorded at the nearest meteorological observatory over the last decade are 14.0–15.7 °C, with an annual average precipitation of 1021–1866 mm

(overall average 1531 mm year$^{-1}$). About half of the annual precipitation occurs during the rainy season from July–September.

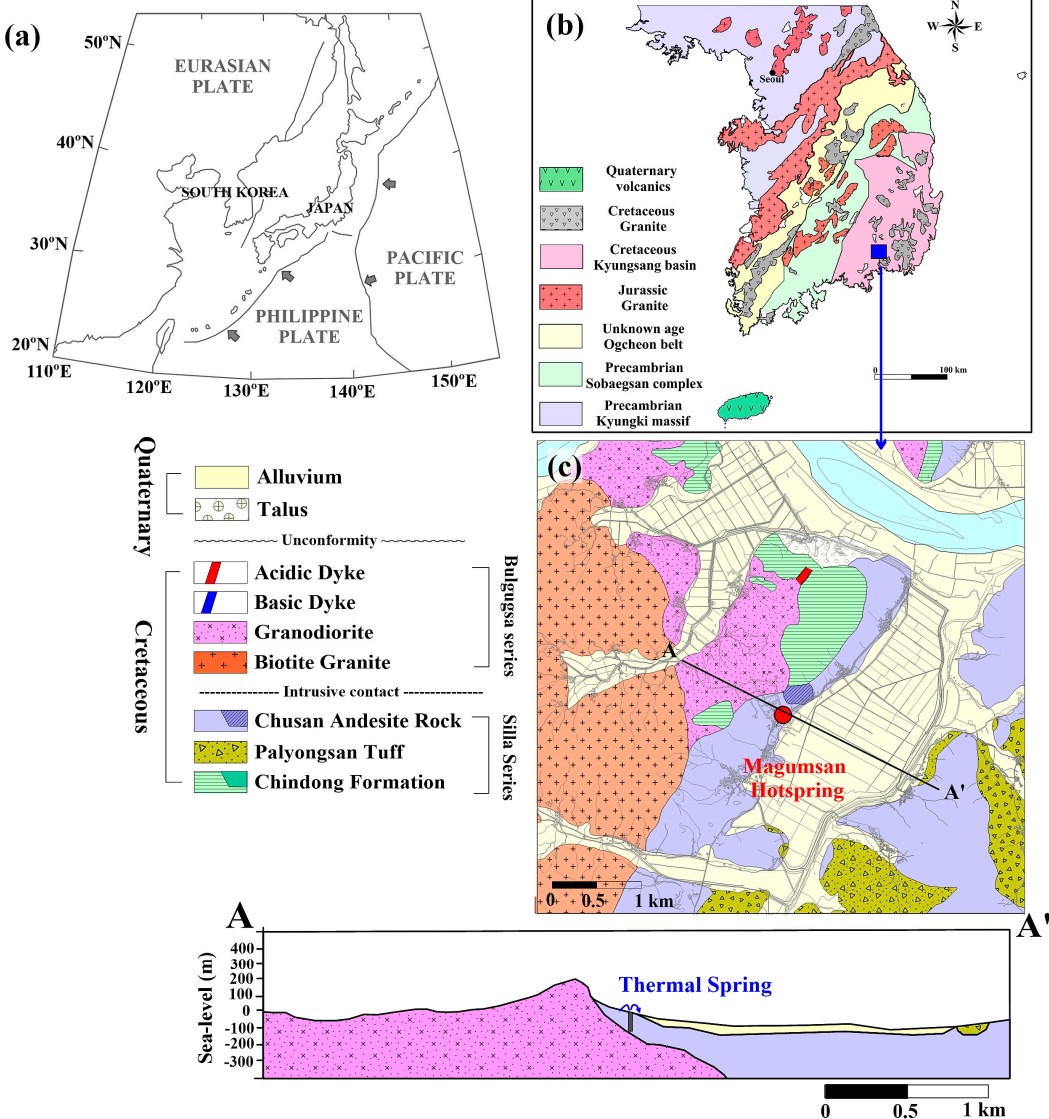

**Figure 1.** Location and geological maps. (**a**) Tectonic framework of northeast Asia; (**b**) major geological units of South Korea; (**c**) geology of the Magumsan area, including a cross-section (A–A') [16].

The Magumsan thermal water arises from andesitic rocks in the Cretaceous Gyeongsang basin, which is a non-marine environment. The geology of the Magumsan area mainly comprises Chusan andesite, which is intruded by granodiorite and biotite granite in the northwest of the study area (Figure 1c). The Chusan andesitic rocks include andesite, trachyte–andesite, and brecciated andesite that occur variously as lava flows, intruded rock, and basement rock. The andesite is highly altered and silicified by hydrothermal reactions with intruding granitic rocks. A drill core in the thermal area has revealed silicified chert as a secondary mineral that results from hydrothermal alteration in the host andesite and it occurs as a thick layer at 170–250 m depth (Figure 2). Calcite veins were likely to have been intercalated before or after the epithermal stage. Pyrite occurs as a secondary mineral in the core and it is more abundant in the deeper parts of the core. Pb and Zn were also found in the core at 125 m depth. An epithermal ore deposit bearing Cu, Pb, and Zn as major ores is located 700 m northwest of the thermal area [1].

Granodiorite in the study area predominantly consists of the oligoclase–labradorite, amphibole, quartz, and biotite. The Chindong Formation, which mainly comprises dark-gray shale and mud rock, is distributed in the northeastern part of the Magumsan thermal area and it is locally intercalated with greenish-gray and dark-brown shales altered to chert and hornfels. The shale and mud rock are composed principally of quartz, feldspar, and fine biotite. Some feldspars are altered to chlorite as a result of hydrothermal alteration by intruding granite [1]. Faults in the study area trend northwest and they dip steeply across the thermal area, and they likely play a role as circulating conduits for thermal water [1,16].

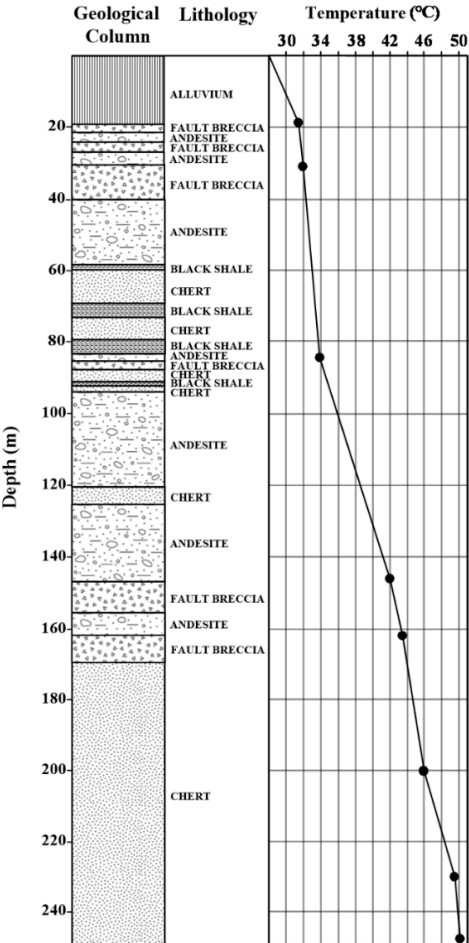

**Figure 2.** Lithological log and temperature gradient in a drilled core from the Magumsan thermal area [1].

## 3. Analytical Methods

### 3.1. Sample Collection and in Situ Measurements

Magumsan thermal waters are pumped out of wells of 300 m average depth. Six thermal water (>30 °C) and five groundwater samples were collected in the Magumsan area from the sites that are shown in Figure 3.

Measurements of pH, oxidation–reduction potential (ORP), dissolved-oxygen content (DO), electrical conductivity (EC), and temperature of all water samples were made in situ while using portable instruments (Orion three star and five star, Thermo Scientific, Beverly, MA, USA) 10–20 min after purging to achieve stable values. Alkalinity, being expressed as $HCO_3^-$ concentration, was quantified on-site by titration with 0.05 M HCl. All of the water samples were filtered through a 0.45 μm membrane filter and then stored in 60 mL polythene bottles. Samples for cation analysis were acidified to pH <2 with ultrapure $HNO_3$.

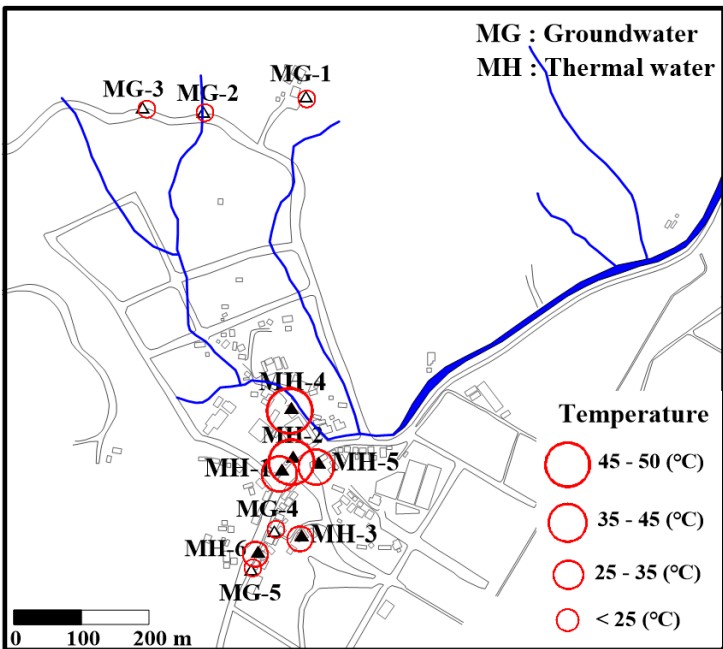

**Figure 3.** Map showing sample locations and temperatures of thermal water (MH) and groundwater (MG) in the Magumsan area.

### 3.2. Chemical and Stable-Isotope Analyses

The concentrations of major cations and minor elements were determined by atomic absorption spectrometry (AAS; Unicam model 989, Unicam Sstemas Analiticos, Lda, Lisbon, Portugal), inductively coupled plasma–atomic emission spectrometry (ICP–AES; Shimadzu model ICPS-1000 III, Kyoto, Japan), and inductively coupled plasma–mass spectrometry (ICP–MS; Fison model PQ III, Thermo Scientific, Massachusetts, MA, USA) at the Korea Basic Science Institute (KBSI) of Daejeon, South Korea. The anion concentrations were determined by ion chromatography (Dionex DX-120 resin, Dionex Corporation, California, CA, USA). The reliability of chemical analysis was estimated by calculating the charge imbalance between major cations and anions, with results of +4.09% to −5.74% for all of the samples.

$\delta^{18}$O, $\delta^2$H, and $\delta^{34}$S values of water samples were determined by stable isotope ratio–mass spectrometry (SIR–MS; Isotoprime model, GV Instruments, Manchester, UK). All of the isotopic analyses were carried out at the KBSI. O and H isotopic ratios are expressed relative to Vienna Standard Mean Ocean Water (V-SMOW), with analytical precisions that were within ±0.1% and ±1.0%, respectively. The determination of sulfate $\delta^{34}$S values involved scavenging by precipitation of $BaSO_4$ after the addition of $BaCl_2$ powder to the thermal waters. Sulfur isotopic ratios are expressed relative to Canon Diablo Troilite (CDT), with analytical precisions within ±0.2%.

### 3.3. Noble-Gas Analysis

The water samples for noble-gas analysis were collected in 50 $cm^3$ vacuum-tight glass bottles with high-vacuum stopcocks on both sides. Analyses were performed at the University of Tokyo, Japan. Noble gases that were dissolved in the water samples were extracted by an all-metal Toepler pump system, which enabled analyses to be performed under ultra-low blank conditions. The extracted gases were compressed into small volumes that were connected to the noble-gas purification line of the MS system. The noble gases were purified by two Ti–Zr getters and then separated into three fractions, He, Ne, and Ar, by charcoal traps and a cryogenically cooled sintered-stainless-steel trap. He, Ne, and Ar isotopic ratios and absolute abundances were determined while using a modified noble-gas mass spectrometer (VG5400, MS-III, Micromass Communication, North Carolina, NC, USA) at the Laboratory for Earthquake Chemistry, University of Tokyo, Japan. The sensitivities and mass

discrimination correction factors were determined by analysis of known amounts of atmosphere using the same procedure as that applied to samples (details are described by [17]). Here, only He and Ne isotopic ratios were considered due to the atmospheric origin of Ar.

## 4. Results and Discussion

### 4.1. Geochemical Compositions

Table 1 provides the in situ measurement data and chemical compositions of thermal waters and groundwaters analyzed in this study.

The temperatures of thermal waters and groundwaters were 30.8–49.8 °C and 17.9–23.5 °C, respectively. pH values of thermal waters were 7.42–7.80, which is slightly higher than those of the groundwaters (5.86–7.35). ORP values of thermal waters indicate reducing conditions with values of −71 to −108 mV (cf. groundwater: +150 to +402 mV). Average EC values of thermal waters were higher than those of groundwaters.

**Table 1.** Geochemical and isotopic compositions of thermal water and groundwater samples that were collected in the Magumsan area.

| Sample ID | Temp (°C) | pH | ORP (mV) | EC (μS cm$^{-1}$) | DO | Na$^+$ | K$^+$ | Ca$^{2+}$ | Mg$^{2+}$ | Sr$^{2+}$ | Fe |
|---|---|---|---|---|---|---|---|---|---|---|---|
| | | | | | | | | | | | |
| MH-1 | 42.2 | 7.78 | −104 | 1272 | 3.40 | 188 | 8.96 | 44.2 | 0.67 | 0.83 | 0.008 |
| MH-2 | 45.0 | 7.52 | −71 | 1044 | 1.30 | 154 | 7.84 | 39.0 | 0.91 | 0.65 | 0.003 |
| MH-3 | 31.3 | 7.48 | −88 | 551 | 1.60 | 92.1 | 2.16 | 20.9 | 0.55 | 0.32 | 0.01 |
| MH-4 | 49.8 | 7.8 | −82 | 1489 | 6.40 | 207 | 13.3 | 47.1 | 0.52 | 1.07 | 0.013 |
| MH-5 | 39.4 | 7.53 | −75 | 1038 | 0.50 | 163 | 6.69 | 33.9 | 0.56 | 0.60 | 0.004 |
| MH-6 | 30.8 | 7.42 | −108 | 740 | 0.10 | 107 | 2.30 | 34.6 | 0.51 | 0.68 | 0.006 |
| MG-1 | 20.8 | 7.35 | +190 | 222 | 4.00 | 14.1 | 0.60 | 26.6 | 4.13 | 0.13 | 0.002 |
| MG-2 | 20.9 | 7.09 | +228 | 188 | 4.90 | 10.1 | 0.61 | 24.6 | 3.66 | 0.90 | 0.117 |
| MG-3 | 23.5 | 7.33 | +402 | 436 | 4.80 | 47.2 | 8.57 | 27.2 | 12.9 | 0.26 | 0.002 |
| MG-4 | 17.9 | 5.86 | +150 | 480 | 1.90 | 39.5 | 5.09 | 38.6 | 11.0 | 0.34 | 0.002 |
| MG-5 | 21.0 | 7.20 | +161 | 664 | 3.10 | 56.4 | 0.93 | 85.1 | 3.78 | 1.85 | 0.002 |

| Sample ID | Mn | SiO$_2$ | HCO$_3^-$ | F$^{2212}$ | Cl$^-$ | SO$_4^{2-}$ | E (%) | $\delta^{18}$O (%) | $\delta^2$H (%) | $\delta^{34}$S (%) |
|---|---|---|---|---|---|---|---|---|---|---|
| MH-1 | 0.61 | 55.8 | 219 | 1.92 | 177 | 57.6 | 3.82 | −8.4 | −55 | 13.0 |
| MH-2 | 10.2 | 54.1 | 214 | 0.17 | 131 | 49.6 | 3.94 | −8.4 | −54 | 10.0 |
| MH-3 | 4.01 | 35.1 | 230 | 0.38 | 24.0 | 19.5 | 2.76 | −7.9 | −51 | 7.0 |
| MH-4 | 8.85 | 62.5 | 186 | 0.48 | 240 | 61.3 | 2.70 | −8.6 | −55 | 15.0 |
| MH-5 | 2.49 | 50.5 | 230 | 0.26 | 143 | 64.0 | −1.26 | −8.2 | −53 | 9.9 |
| MH-6 | 18.4 | 34.4 | 307 | 0.36 | 35.5 | 24.7 | −0.65 | −8.2 | −52 | 7.2 |
| MG-1 | 0.002 | 33.2 | 134 | <0.05 | 7.05 | 6.88 | −5.81 | −8.4 | −57 | |
| MG-2 | 0.002 | 33.2 | 118 | <0.05 | 3.13 | 4.89 | −5.18 | −8.3 | −57 | |
| MG-3 | 0.002 | 22.9 | 177 | <0.05 | 21.6 | 37.2 | 4.08 | −8.1 | −58 | |
| MG-4 | 0.002 | 18.0 | 126 | <0.05 | 50.6 | 21.8 | −3.58 | −7.6 | −55 | |
| MG-5 | 0.005 | 26.9 | 328 | <0.05 | 28.7 | 67.8 | −4.54 | −7.8 | −54 | |

Notes: E: charge balance error.

Major-ion contents (meq L$^{-1}$) of thermal waters (MH) and groundwaters (MG) are compared in a Schöeller diagram (Figure 4). Na$^+$, K$^+$, SO$_4^{2-}$, and Cl$^-$ generally have higher concentration ranges in thermal waters than in groundwater, whereas Mg and Mn contents display more striking differences: the Mg content of the thermal waters is very low (<1 mg L$^{-1}$) when compared with that of groundwaters (up to 12.9 mg L$^{-1}$), and the Mn content of groundwater (<5 μg L$^{-1}$) is much lower than that of the thermal waters (up to 18.4 mg L$^{-1}$). The low Mg content of thermal waters can be explained by the absorption of Mg in high-temperature geothermal systems. As the geothermal fluids flow from a high temperature to a low temperature environment, it appears to adsorb large amounts of Mg into

contacting host rocks [13,18]. The occurrence of Mn in thermal water might be due to the dissolution of secondary Mn-bearing minerals that are associated with epithermal activity.

Figure 5 indicates the influence of temperature on pH, EC, and major-ion concentrations in thermal waters and groundwaters, in which the correlation coefficients (R) between variables are given. All of the parameters, except $HCO_3^-$ concentration in thermal waters, are positively correlated with temperature with $R^2$ values of 0.59–0.89, indicating that temperature plays an important role in the promotion of water–rock interactions. The inverse trend of $HCO_3^-$ in thermal waters may be explained by increasing temperature causing a decrease in the dissolution rate of $CO_2$. In Figure 5, $Na^+ + K^+$, $SO_4^{2-}$, $Cl^-$, and $SiO_2$ concentrations in thermal water as a function of temperature can be separated into two types: (1) A type of a higher-ion concentration at higher temperatures and (2) B type of a lower-ion concentration at lower temperatures, with these types corresponding to chemical types of Piper diagram.

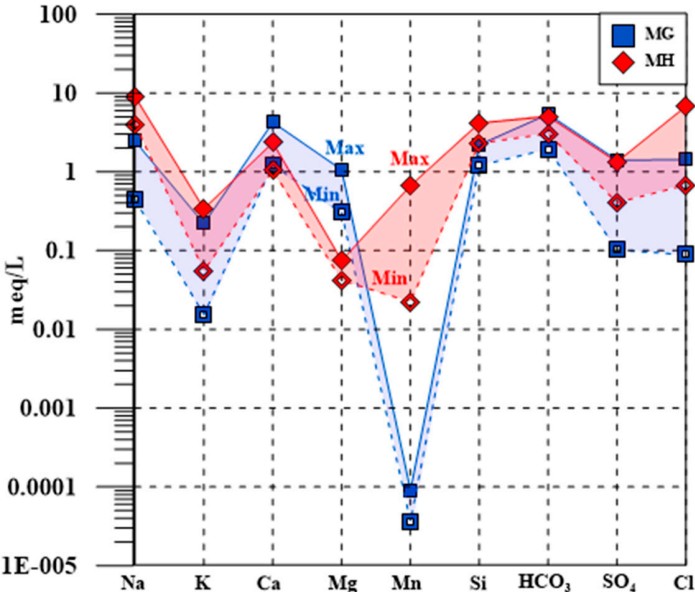

**Figure 4.** Schöeller diagram comparing concentration ranges of major ions in thermal water (MH) and groundwater (MG). Filled and open symbols indicate maximum and minimum values, respectively.

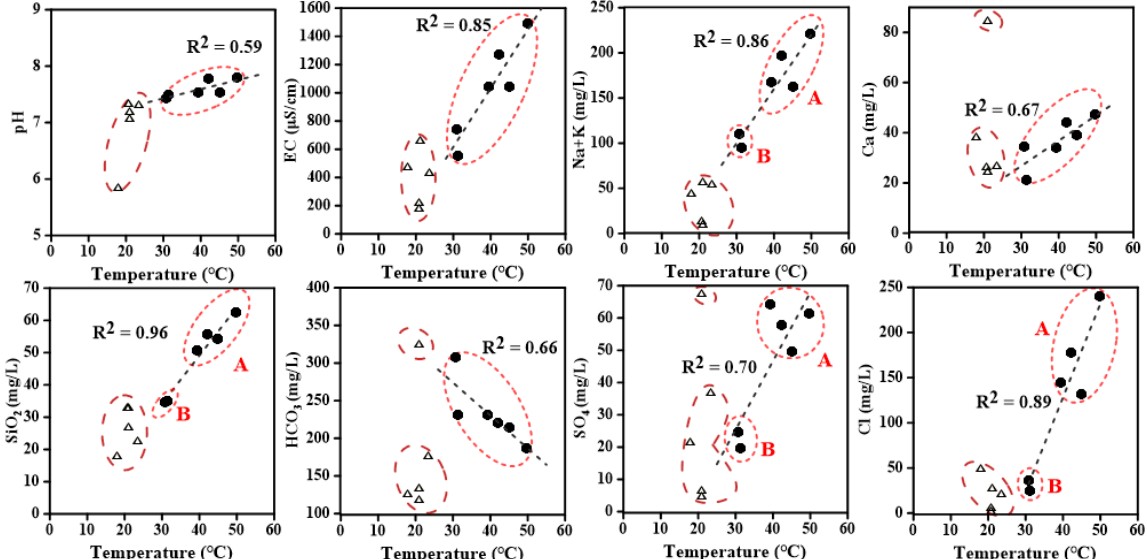

**Figure 5.** Relationships between chemical parameters and the temperatures of thermal waters (filled circles) and groundwater (open triangles).

Major cations and anions (expressed as meq percentiles) are shown in a trilinear plot to delineate the geochemical evolution and chemical types of thermal waters and groundwaters (Piper diagram; Figure 6). Chemical compositions of the groundwaters (MG) are of the Ca–HCO$_3$ type, whereas the thermal waters (MH) can be chemically divided into a Na–HCO$_3$ type at lower temperatures (type I) and an Na–Cl (HCO$_3^-$, SO$_4^{2-}$) type at higher temperatures (type II). It is generally considered that, in the geochemical evolution of thermal waters and deep groundwater in granite and gneiss areas of South Korea, the waters are of the Ca–HCO$_3$ type initially, progressing through the Ca(Na)–HCO$_3$ type to the alkaline Na–HCO$_3$ type in the final stage [13,19]. The two different types of thermal water appear to have evolved in different geological environments.

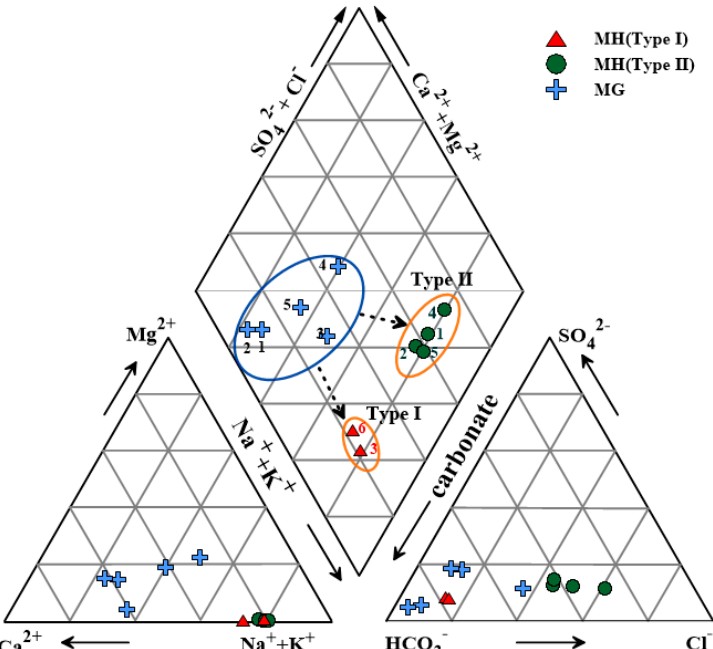

**Figure 6.** Piper diagram of the major chemical compositions of thermal waters (MH: filled triangles and circles) and groundwater (MG: blue pluses) expressed in milli-equivalent percentages. See Section 4.1 for details.

### 4.2. O, H, and S Isotopic Compositions

Table 1 shows the oxygen ($\delta^{18}$O) and hydrogen ($\delta^2$H) isotopic compositions of thermal waters and local groundwater. The $\delta^{18}$O and $\delta^2$H values of thermal water are in the ranges of −8.6% to −7.9%, and −55% to −51%, respectively. The $\delta^{18}$O and $\delta^2$H values of the local groundwater are in the ranges of −8.4% to −7.6% and −58% to −54%, respectively.

In a $\delta^2$H–$\delta^{18}$O diagram (Figure 7), $\delta^{18}$O and $\delta^2$H values plot close to the global meteoric water line (GMWL) [20] and local meteoric water line (LMWL), with thermal waters above and groundwaters below or on the GMWL/LMWL, with the following local isotopic relationship applying: $\delta^2$D = 7.78$\delta^{18}$O + 5.3 [21]. The isotopic fractionation of thermal waters after meteoric water recharge results in D enrichment, with such an enrichment in other thermal waters of South Korea having been previously reported [2]. It may be explained that the clustering of thermal waters above the LMWL that indicates deuterium enrichment is a result of H$_2$S exchange [22,23]. H$_2$S isotopic exchange reaction, which may lead to a significant deviation from the initial stable isotope content of thermal water, is the following: $^2$H$^1$HS($g$) + $^1$H$^1$HO($l$) = $^1$H$^1$HS($g$) + $^2$H$^1$HO($l$) [23].

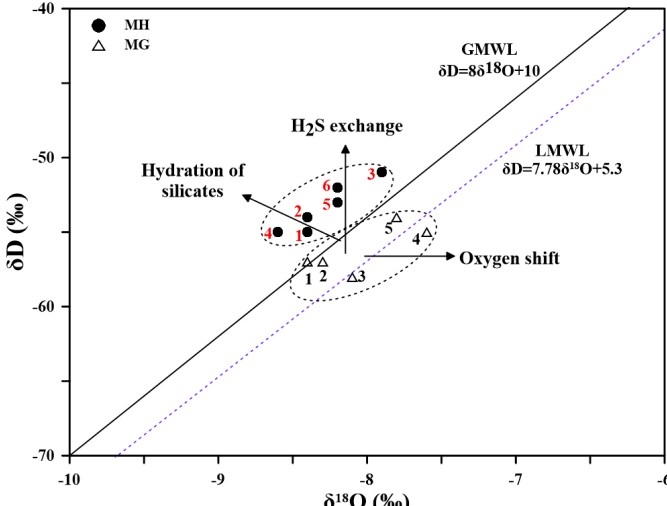

**Figure 7.** $\delta^{18}O$–$\delta D$ relationship for thermal waters and groundwaters in the study area. Numbers indicate the sample number of thermal waters (MH) and groundwaters (MG). (GMWL: Global Meteoric Water Line; LMWL: Local Meteoric Water Line).

The oxygen shift due to oxygen isotopic exchange in water–rock interactions, as observed in high-temperature thermal waters of Japan and other countries [24], is not usually observed in the Magumsan thermal waters. Although isotopic fractionation in thermal waters occurs in a narrow range (within 1% for $\delta^{18}O$), the $\delta^{18}O$ and $\delta D$ values of thermal waters are negatively correlated with temperature ($R^2$ = 0.79–0.82) (Figure 8), with higher-temperature thermal waters having more isotopically depleted values. This may reflects the altitude effect, with the more depleted water being recharged at higher altitudes and migrating deeper into the heat source, and also reflecting another effect, such as silicate hydration. During the hydration of silicate minerals in thermal water contact with rocks, the fractured rocks would preferentially use heavy isotopes, which cause the fractionation of thermal water enriched in light isotopes [25].

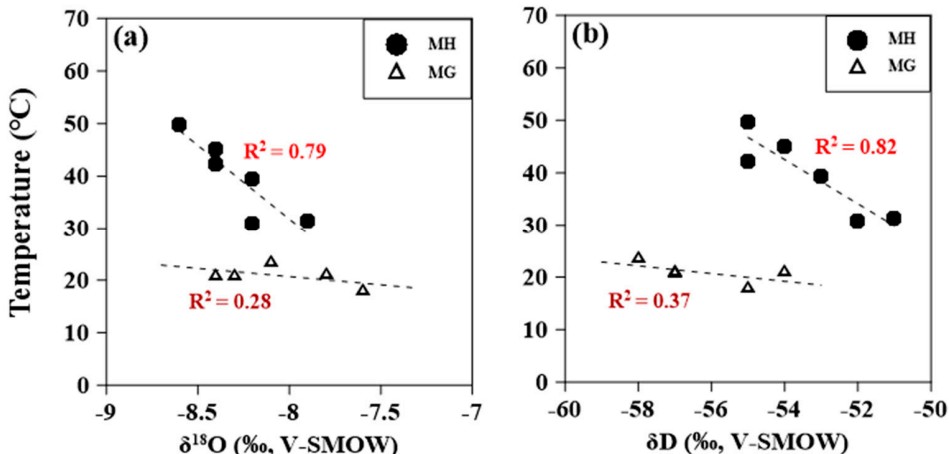

**Figure 8.** Relationships between temperature and $\delta^{18}O$ (**a**) and $\delta D$ (**b**) of the thermal waters (MH) and groundwaters (MG).

The sulfur isotopic ratios ($\delta^{34}S$) were determined to elucidate the origin of sulfur dissolved as sulfate in Magumsan thermal waters. The $\delta^{34}S$ values of dissolved sulfate ranged from +7% to +15% (Table 1), with the thermal waters being classified into two types A type of a lower level of +7.0% to +7.2%, and B type of a higher level of +9.9% to +15.0% (Figure 9). This classification corresponds to the chemical types of thermal waters in the trilinear plot of Figure 6, with the former

type representing the lower-temperature Na–HCO$_3$ type, and the latter type the higher-temperature Na–Cl (HCO$_3$, SO$_4$) type.

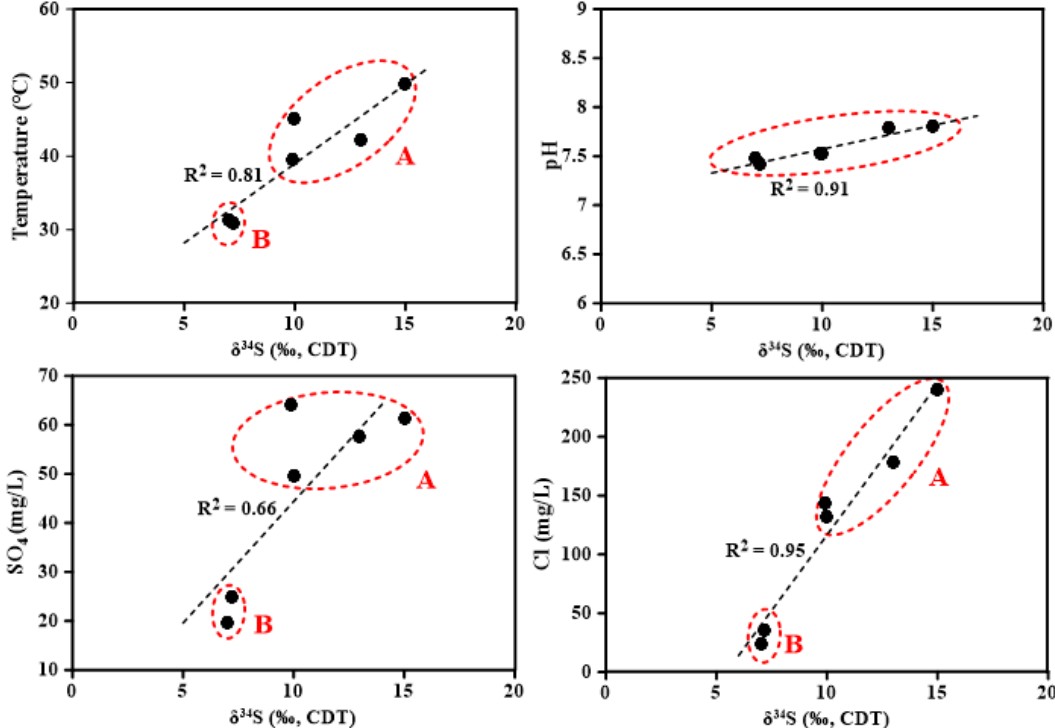

**Figure 9.** Relationships between chemical composition, temperature, and δ$^{34}$S of the thermal water.

The δ$^{34}$S values of sulfide minerals in ore deposits of South Korea are known to have ranges of −2.1% to 9.2% [25–28], whereas the δ$^{34}$S values of atmospheric and seawater sulfur have ranges of +6% to +7% and values of +20%, respectively [29]. Kim and Nakai [2] reported δ$^{34}$S values of dissolved sulfate in 17 hot-spring waters in South Korea in the range of +5.5% to +29.3%, and classified the thermal waters into three groups: (1) those including sulfate of seawater origin (δ$^{34}$S > +19%), (2) those with sulfate formed by oxidation of igneous sulfide minerals (δ$^{34}$S +5% to +10%), and (3) those with a mixture of these sources (δ$^{34}$S +10% to 19%). Although only one Magumsan thermal water sample (at 40.7 °C) was analyzed by Kim and Nakai [2], those authors suggested that sulfate S originated from mixed sources of seawater and magmatic pyrite sulfate.

Pyrite is commonly found in andesitic rocks and drill cores, and it is generally considered that the oxidation of pyrite is a source of sulfate in low-δ$^{34}$S (<7.2%) thermal waters. Here, we consider that there are two possible sources of higher δ$^{34}$S (>9.9%) values: (1) the derivation of sulfate from the kinetic oxidation of magmatic pyrites in the thermal aquifer system, with kinetically controlled chemical oxidation of reduced sulfides (and sulfur) forming SO$_4^{2-}$ that was enriched in $^{34}$S [30,31]; and, (2) paleo-seawater mixing, as indicated by the positive correlation ($R^2$ = 0.66–0.95) between temperature, Cl$^-$, SO$_4^{2-}$, and δ$^{34}$S in thermal waters (Figure 9). The δ$^{34}$S values are classified into two distinct groups (higher and lower) in terms of temperature and SO$_4^{2-}$ and Cl$^-$ concentrations in thermal waters.

## 4.3. Helium Analysis

The isotopes of noble gases are useful for studying heat sources [32,33]. In thermal systems, the atmospheric $^3$He/$^4$He ratio (Ra = 1.384 × 10$^{-6}$) is generally used as a reference value [34]. He isotopic ratios are generally interpreted in terms of the mixing of a MORB-type upper-mantle source end-member (average $^3$He/$^4$He = 12 × 10$^{-6}$, or 8.5 ± 1 Ra [35]) and a crustal source end-member ($^3$He/$^4$He 0.01 ×

$10^{-6}$, or 0.01–0.1 Ra [36]), where $^3$He is mantle derived and $^4$He is predominantly of crustal origin and associated with radioactive decay of U–Th series elements.

Here, the He and Ne isotopic ratios were studied in the Magumsan thermal waters, with Table 2 presenting concentrations and isotopic ratios of noble gases. $^3$He/$^4$He and $^4$He/$^{20}$Ne ratios of thermal waters were in the ranges 0.335–0.442 ($\times 10^{-6}$) and 2.08–21.7, respectively, which indicated the predominance of crustal He in thermal waters.

**Table 2.** Helium and neon isotopic data for Magumsan thermal waters.

| Sample ID | $^4$He ($\times 10^{-7}$ cm$^3$ g$^{-1}$) | $^{20}$Ne ($\times 10^{-9}$ cm$^3$ g$^{-1}$) | $^3$He/$^4$He ($\times 10^{-6}$) | $^4$He/$^{20}$Ne |
|---|---|---|---|---|
| MH-1 | 0.320 | 271 | 0.335 | 11.8 |
| MH-2 | 0.608 | 344 | 0.442 | 17.6 |
| MH-3 | 0.102 | 491 | 0.420 | 2.08 |
| MH-5 | 0.460 | 212 | 0.340 | 21.7 |
| MH-6 | 0.238 | 301 | 0.366 | 7.89 |

In a $^3$He/$^4$He–$^4$He/$^{20}$Ne diagram (Figure 10), the $^4$He/$^{20}$Ne ratio can be considered as an indicator of atmospheric He contamination in the thermal waters, with He ratios for atmospheric, crustal, and upper-mantle end-members being shown for comparison. The air–mantle mixing line represents a mixture of isotopic ratios for atmospheric He and Ne ($^3$He/$^4$He = 1.384 × 10$^{-6}$; $^3$He/$^{20}$Ne = 0.317 [37]) and those of MORB-type mantle origins ($^3$He/$^4$He = 12 × 10$^{-6}$, or 8.5 ± 1 Ra; $^4$He/$^{20}$Ne > 10,000 [35]). The air–crust mixing line represents a mixture of atmospheric and crustal He and Ne ($^3$He/$^4$He = 5 × 10$^{-9}$; $^4$He/$^{20}$Ne > 10,000). Although the crustal $^3$He/$^4$He ratios range from 4 × 10$^{-9}$ to 1 × 10$^{-8}$, depending on the chemical composition of the crust [36], a crustal $^3$He/$^4$He ratio of 5 × 10$^{-9}$ remains valid for the entire range of crustal values when being expressed in terms of orders of magnitude.

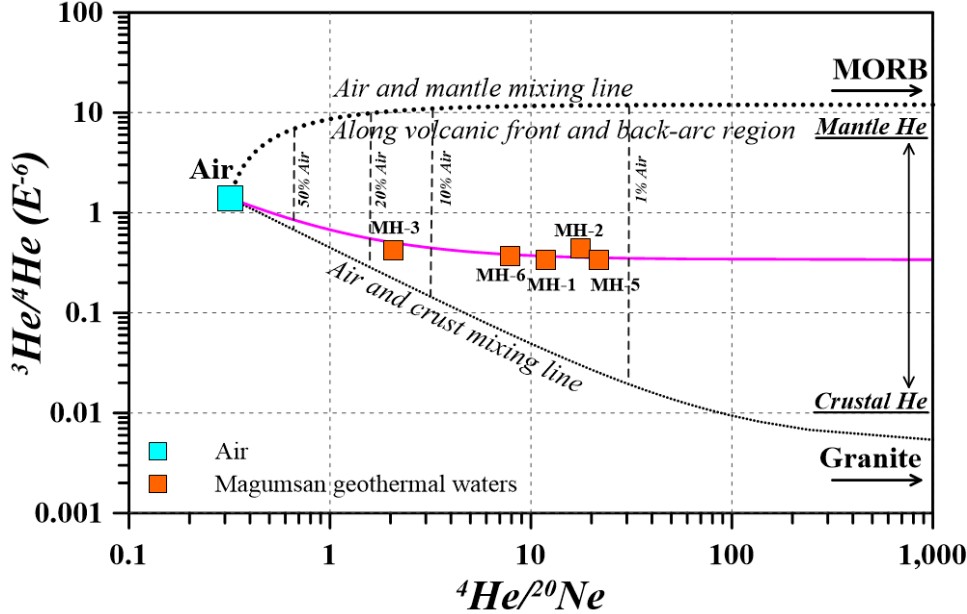

**Figure 10.** Three-end-member mixing diagram for He isotopic ratios in Magumsan thermal waters. (MH-3: 15.3% air contaminated sample; other samples: 1.5%–4.0% air contaminated samples).

The mixing line that passes through the data points (Figure 10) was calculated by assuming two-component mixing between atmospheric He and Ne and a source with unknown $^3$He/$^4$He ratio and $^4$He/$^{20}$Ne = 10,000 [38]. Park et al. (2016) provided details of the calculation [9].

Notably, He data for thermal waters plot on a single air-mixing line between mantle and crustal He components (Figure 10) with $^3$He/$^4$He = 0.34 × 10$^{-6}$, which is a similar ratio to that observed in

other thermal waters in South Korea not enriched in mantle [3]He [6–9]. The distribution of He isotopic ratios on a single mixing line also indicates that He originates from a common source, with the range of $^4$He/$^{20}$Ne ratios representing the degree of atmospheric He contamination through circulating meteoric thermal water that is saturated with dissolved noble gases.

The measured He concentrations in thermal waters represent a mixture of three source components (atmosphere, crust, and mantle), with mixing ratios, as given for $^3$He and $^4$He in Table 3. These ratios were calculated while using the method of Kotarba and Nagao (2008) [39]. In thermal waters, the crust-derived $^4$He contributed 83%–96% and mantle-derived $^4$He 1.6%–3.4% of the total $^4$He, with the atmospheric contribution being 1.5%–15.3%. The mantle- and crust-derived $^3$He contributions were in the ranges 47%–92.2% and 2.0%–2.8%, respectively. Sample MH-3 has a high $^3$He air-contamination ratio of 51% (15.3% for $^4$He).

**Table 3.** Mixing ratios of three sources of $^3$He and $^4$He in Magumsan thermal waters.

| Sample No. | $^4$He | | | $^3$He | | |
|---|---|---|---|---|---|---|
| | Air (%) | Mantle (%) | Crust (%) | Air (%) | Mantle (%) | Crust (%) |
| MH-1 | 2.7 | 2.4 | 94.9 | 11.3 | 85.9 | 2.8 |
| MH-2 | 1.8 | 3.4 | 94.8 | 5.7 | 92.2 | 2.1 |
| MH-3 | 15.3 | 1.6 | 83.0 | 51.0 | 47.0 | 2.0 |
| MH-5 | 1.5 | 2.6 | 96.0 | 6.0 | 91.1 | 2.8 |
| MH-6 | 4.0 | 2.5 | 93.5 | 15.4 | 82.1 | 2.6 |

Figure 11 shows the relationships between the $^4$He concentrations and $^4$He/$^{20}$Ne ratios and temperature, where both are positively correlated with temperature ($R^2$ = 0.72 and 0.57, respectively). These relationships indicate that the heat source for Magumsan thermal waters is strongly associated with the radioactive decay of U and Th in the crustal rocks.

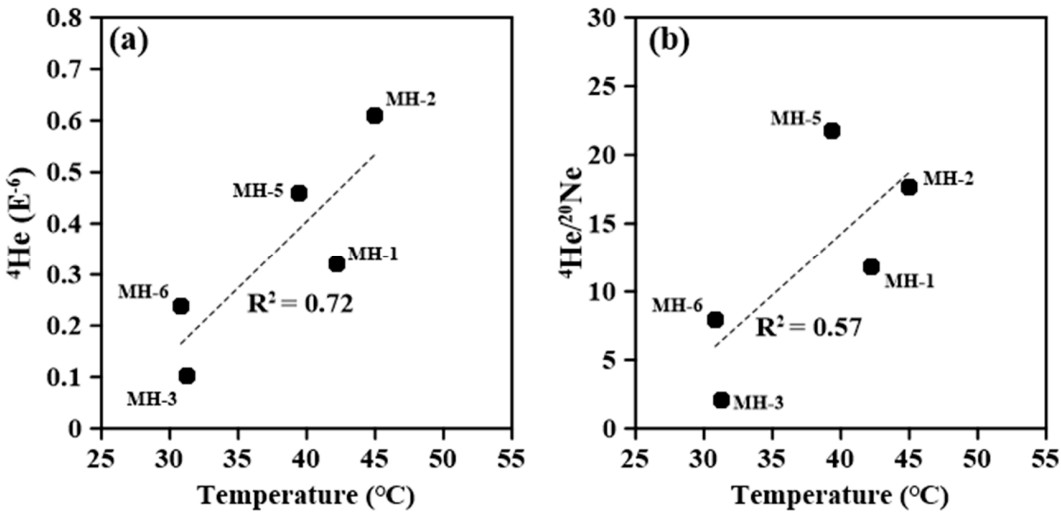

**Figure 11.** Correlations between $^4$He concentration (**a**) and $^4$He/$^{20}$Ne isotopic ratio (**b**) and temperature of thermal waters.

### 4.4. Geothermometry and a Mixing Model for the Thermal Reservoir

Cation and silica geothermometers are often used to estimate the temperatures of thermal waters. A requirement for the successful application of cation geothermometers is the attainment of water–rock chemical equilibrium in the geothermal reservoir [13]. The mixing of thermal and cold waters may cause errors in chemical geothermometry, but silica geothermometry can provide realistic estimates of reservoir temperature, even over the ranges of 120–180 °C [40]. Chalcedony geothermometry can also be utilized at temperatures of <190 °C [13,41,42].

Table 4 presents Magumsan reservoir temperatures that are estimated using cation, quartz, and chalcedony geothermometers. Cation geothermometry results show little agreement, with uncertainties mainly resulting from the mixing of thermal and shallow cold water, and a lack of equilibrium [43]. Each geothermometer has its own limitations [44,45], and water–rock equilibrium in geothermal reservoirs is a prerequisite for accuracy. Even so, it is necessary to select an appropriate method that is based on the chemical characteristics of the thermal water as determined by a Na–K–Mg ternary diagram (Figure 12) [10].

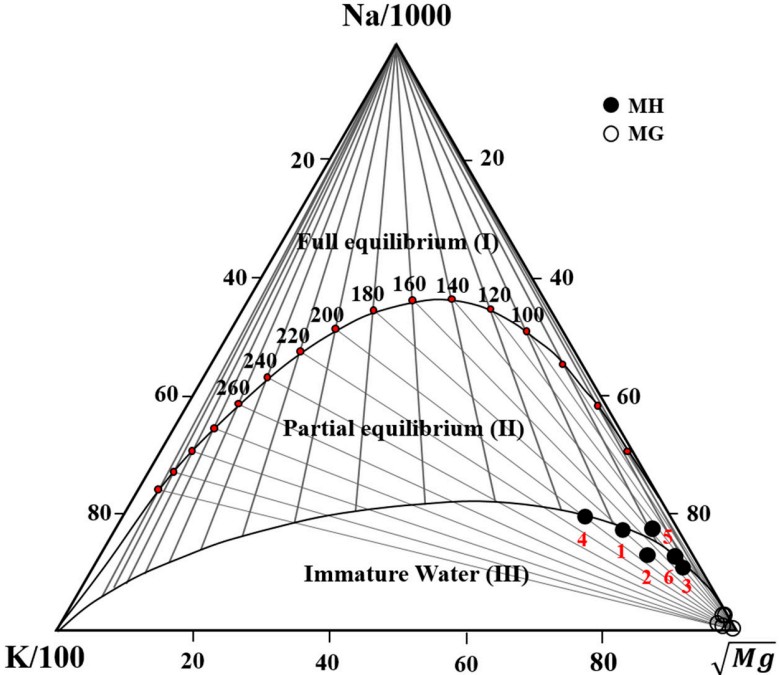

**Figure 12.** Ternary Na–K–Mg plot of thermal waters (filled circles with number) and groundwaters (open circles), showing water–rock equilibrium temperatures [11].

The Na–K–Mg ternary diagram has been previously used to estimate the reservoir temperatures and to gauge whether equilibrium has been attained between thermal water and mineral assemblages comprising albite, K-feldspar, muscovite, and clinochlore [11]. The diagram classifies water into three groups (zones): (I) fully equilibrated water (mature), (II) partially equilibrated water (mixed water), and (III) immature waters (Figure 12). For waters within zone III, where water–rock equilibrium does not exist, cation geothermometry would not provide reliable results [46]. The thermal waters and groundwaters in the study area plot along (or around) the boundary between zones II and III, and in zone III (Figure 12). The thermal waters are most likely on the boundary between partial and no equilibrium due to their ascent from a deep aquifer and mixing with cold shallow waters at certain ratios. This mixing causes changes in both temperature and chemical composition, and full water–rock equilibrium may not be attained [13,18].

Silica (quartz and chalcedony) and cation (Na–K–Ca and Na–K) geothermometers were used to estimate the reservoir temperatures of the Magumsan geothermal system while using the chemical analyses of the thermal waters using the equations that are listed in Table 4.

Reservoir temperatures of thermal waters that are calculated by quartz and chalcedony geothermometers are in the ranges 85–113 °C and 54–86 °C, respectively. Thermal/cold water mixing ratios that are calculated by these geothermometers are 68%–85% and 54%–74%, respectively (Table 5). Therefore, the silica geothermometer is considered to be the most suitable for application in the Magumsan area, and the silica–enthalpy model [47] was applied. In this method, enthalpy is used

as a coordinate rather than temperature, because the combined heat contents of two waters at different temperatures are conserved when the waters are mixed but their combined temperatures are not [12].

**Table 4.** Mathematical equations to estimate reservoir temperatures.

| Geothermometers | Equations | Source |
|---|---|---|
| Qtz [a] (°C) | $T = \dfrac{1309}{5.19 - log\mathrm{SiO_2}} - 273.15$ | [42] |
| Qtz [b] (°C) | $T = \dfrac{1522}{5.75 - log\mathrm{SiO_2}} - 273.15$ | [42] |
| Chal [a] (°C) | $T = \dfrac{1309}{4.69 - log\mathrm{SiO_2}} - 273.15$ | [42] |
| Chal [b] (°C) | $T = \dfrac{1309}{5.09 - log\mathrm{SiO_2}} - 273.15$ | [42] |
| Na–K–Ca (°C) | $T = \dfrac{1647}{\left(log\dfrac{Na}{K} + \beta log\dfrac{Ca^{0.5}}{Na} + 2.06\right) + 2.47} - 273.15, \ \beta = \dfrac{4}{3} \ for \ T < 100°C$ | [48] |
| Na–K[1] (°C) | $T = \dfrac{856}{0.857 + log\dfrac{Na}{K}} - 273.15$ | [49] |
| Na–K[2] (°C) | $T = \dfrac{1217}{1.483 + log\dfrac{Na}{K}} - 273.15$ | [50] |
| Na–K[3] (°C) | $T = \dfrac{933}{0.933 + log\dfrac{Na}{K}} - 273.15$ | [51] |
| K–Mg (°C) | $T = \dfrac{4410}{14.0 - log\dfrac{Na}{K}} - 273.15$ | [11] |

Note: [a] no steam loss; [b] maximum steam loss (at 100 °C); Na–K[1], Na–K[2] and Na–K[3] are the equations suggested by Truesdell, Fournier and Arnórsson, respectively; T is the model reservoir temperature in °C; S is the silica concentration of the thermal water in mg/L; and Na, K, Ca and Mg concentrations are expressed in mol/L.

**Table 5.** Estimated reservoir temperatures and mixing ratios of thermal waters using quartz, chalcedony, and cation geothermometers.

| Sample ID. | Estimated Reservoir Temperature and Mixing Ratio | | | | | | | | | | |
|---|---|---|---|---|---|---|---|---|---|---|---|
| | Outflow (°C) | Qtz [a] (°C) | Qtz [b] (°C) | Mix [q] (%) | Chal [a] (°C) | Chal [b] (°C) | Mix [ch] (%) | Na–K–Ca (°C) | Na–K[1] (°C) | Na–K[2] (°C) | Na–K[3] (°C) K–Mg (°C) |
| MH-1 | 42.2 | 107 | 107 | 0.75 | 77.5 | 80.4 | 0.64 | 63.3 | 82.1 | 127.8 | 102.3 48.1 |
| MH-2 | 45.0 | 106 | 106 | 0.72 | 75.9 | 79.0 | 0.58 | 59.9 | 86.4 | 131.6 | 106.6 42.4 |
| MH-3 | 31.3 | 86.0 | 88.8 | 0.85 | 55.0 | 60.3 | 0.73 | 31.9 | 41.9 | 90.9 | 60.9 23.3 |
| MH-4 | 49.8 | 113 | 112 | 0.68 | 83.4 | 85.7 | 0.55 | 75.2 | 102.4 | 145.7 | 122.9 59.1 |
| MH-5 | 39.4 | 102.3 | 103 | 0.77 | 72.4 | 75.9 | 0.66 | 42.1 | 40.1 | 89.2 | 59.0 32.7 |
| MH-6 | 30.8 | 85.1 | 88.1 | 0.85 | 54.1 | 59.5 | 0.74 | 26.5 | 37.5 | 86.8 | 56.4 25.1 |

Note: Qtz, Quartz; Chal, Chalcedony; [a] no steam loss; [b] maximum steam loss (at 100 °C); Mix [q] and Mix [ch] are the mixing ratio about quartz and chalcedony, respectively; Na–K[1], Na–K[2] and Na–K[3] are the equations suggested by Truesdell, Fournier and Arnórsson, respectively.

In a silica-enthalpy diagram (Figure 13), the mean temperature of cold shallow water was assumed to equal the annual mean temperature (14.5 °C) in the study area, and the $SiO_2$ concentrations of thermal waters and shallow cold water were assumed to be 45–69 mg L$^{-1}$ and 17.6 mg L$^{-1}$, respectively, based on the data for surface water of Choi and Kim [1]. With reference to the steam table [52], the enthalpies of the thermal waters and cold shallow water were calculated to be 129–208 KJ kg$^{-1}$, and 65 KJ kg$^{-1}$, respectively. The intersection point of the mixing line with the solubility of chalcedony indicates an enthalpy of 555–707 KJ kg$^{-1}$ and a maximum silica content of 63 mg L$^{-1}$ in thermal water. The estimated

reservoir temperature was in the range of 132–167 °C. Table 6 gives the reservoir enthalpies, silica concentrations, and mixing ratios that were calculated while using the silica–enthalpy method.

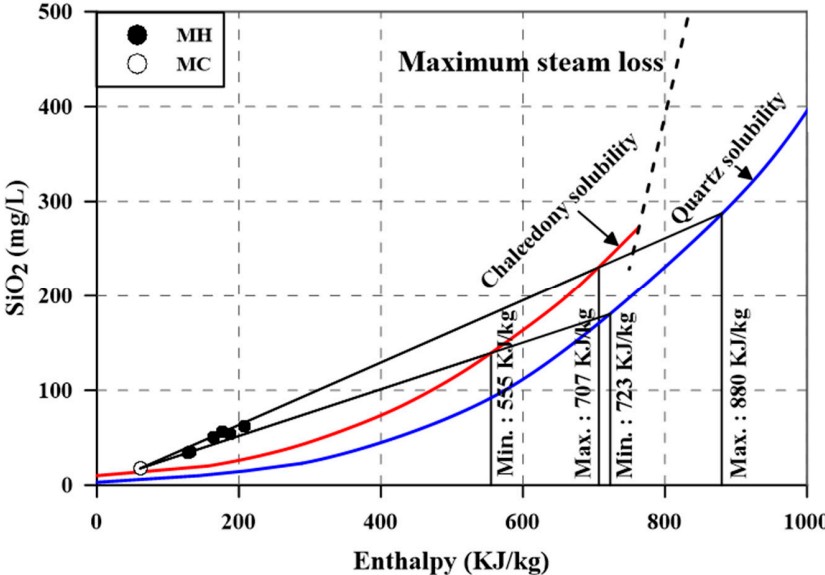

**Figure 13.** Enthalpy–silica mixing diagram of thermal waters (filled circles) and cold shallow water (open circle) in the Magumsan area.

**Table 6.** Estimated reservoir temperatures of thermal waters and their mixing ratio with cold water based on the mixing model of enthalpy and silica.

| Sample ID. | Enthalpy (KJ kg$^{-1}$) | | | Mixing Ratio (%) | SiO$_2$ (mg L$^{-1}$) | | | Mixing Ratio (%) |
|---|---|---|---|---|---|---|---|---|
| | Outflow | Reservoir | Cold Water | | Outflow | Reservoir | Cold Water | |
| MH-1 | 177 | 880 | 62.5 | 0.86 | 55.8 | 279 | 26.8 | 0.89 |
| MH-2 | 188 | 795 | 62.5 | 0.83 | 54.1 | 213 | 26.8 | 0.85 |
| MH-3 | 133 | 726 | 62.5 | 0.89 | 35.1 | 131 | 26.8 | 0.92 |
| MH-4 | 208 | 830 | 62.5 | 0.81 | 62.5 | 241 | 26.8 | 0.83 |
| MH-5 | 165 | 855 | 62.5 | 0.87 | 50.5 | 254 | 26.8 | 0.90 |
| MH-6 | 129 | 723 | 62.5 | 0.90 | 34.4 | 125 | 26.8 | 0.92 |

Note: Enthalpy and SiO$_2$ of shallow cold water are assumed to be 14.5 °C (60.8 KJ kg$^{-1}$) and 26.8 mg L$^{-1}$, respectively.

Regarding quartz solubility, it was assumed that the reservoir temperature is in the range 171–206 °C, which is higher than the range (54–113 °C) that was calculated by chalcedony and quartz geothermometry. The overestimated temperature probably resulted from the occurrence of conductive cooling [12]. The mixing ratio of thermal and cold water can be estimated while using a graphical method with the silica–enthalpy model [51]. The mixing ratio of cold/thermal water was estimated at 76%–86% for chalcedony and 81%–90% for quartz, with these ratios being higher than those that were given by chalcedony geothermometry, which was considered to be a reliable method.

The thermodynamic equilibrium geothermometer that is based on the calculation of saturation index (SI) values in water–mineral interaction systems can also be used to assess thermal reservoir temperatures. Here, SI values for carbonate (calcite and dolomite), silicate minerals (albite, anorthite, K-mica, chlorite, illite, and kaolinite), and silica minerals (quartz and chalcedony), which are potential reactive primary minerals and secondary minerals, were used to predict the tendency for precipitation or the dissolution of these minerals, with SI = 0 indicating thermodynamic equilibrium, SI > 0 indicating oversaturation (precipitation conditions), and SI < 0 indicating undersaturation (dissolution conditions).

The chemical data of Table 1 were used to calculate SI values of thermal waters while using the software program PHREEQC [53,54] under the assumption that no conductive cooling occurs.

The equilibrium state between the minerals and water is temperature dependent. Saturation indices between chemical compositions of thermal waters and potential reactive minerals were calculated as a function of temperature, with the results being shown in Figure 14. The results indicate that the thermal waters are oversaturated with respect to kaolinite, K-mica, calcite, chalcedony, and quartz, and they are undersaturated with respect to albite, anorthite, illite, chlorite, and dolomite at their discharge temperatures. The composition of thermal water is more likely to reflect the state of water–rock equilibrium at deeper reservoir temperatures, rather than at the discharge temperature. This value is most likely the reservoir temperature when the equilibrium lines of a series of potential reactive minerals converge at a certain temperature [55].

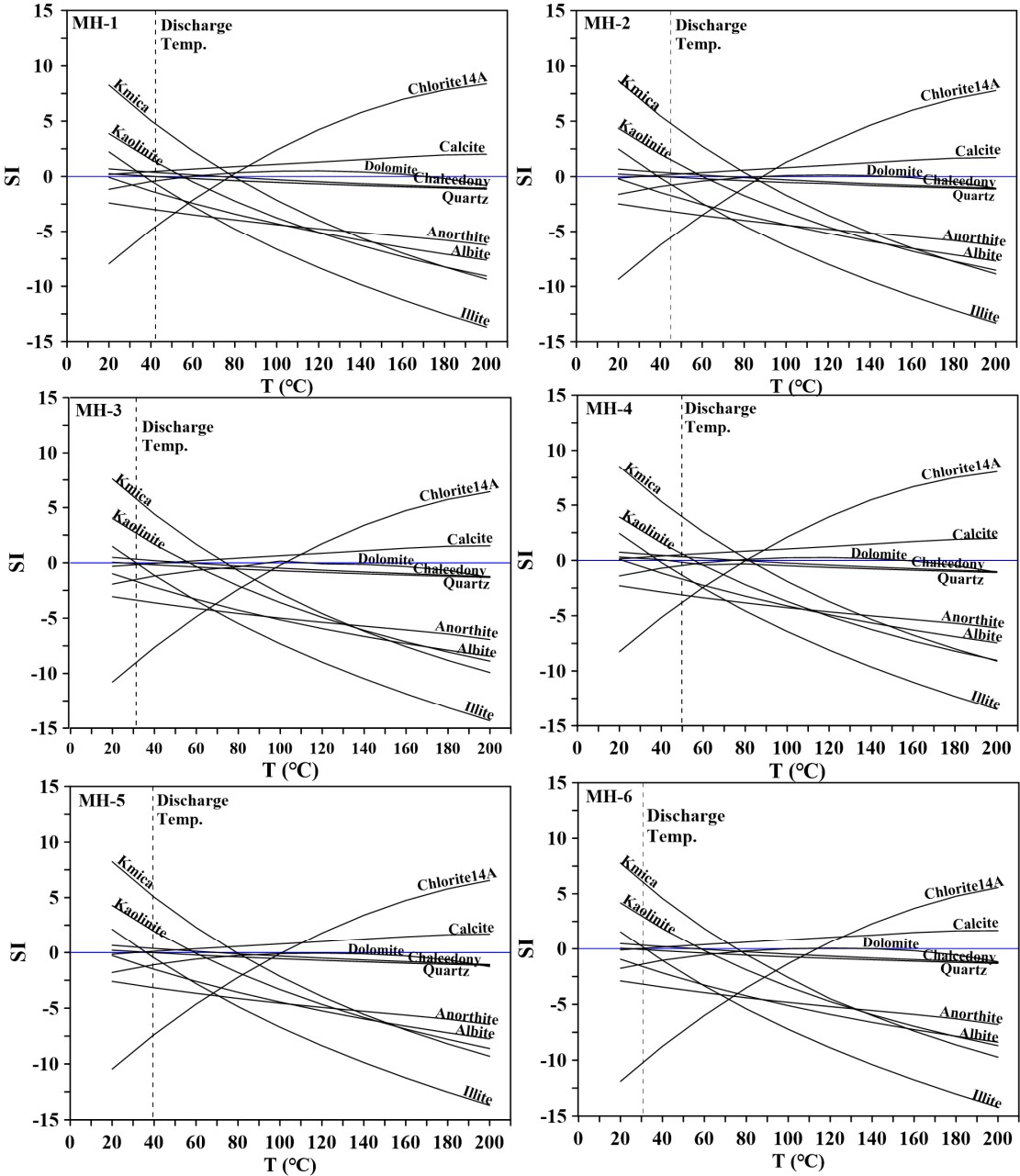

**Figure 14.** Changes in saturation indices of thermal waters with respect to potential reactive minerals, as a function of temperature.

SI values of kaolinite, K-mica, dolomite, chalcedony, and quartz for the thermal waters converge on and lie close to SI = 0 in the temperature range of 55–83 °C. This may be the temperature range in which the maximum number of mineral phases is in equilibrium with the thermal waters, and can thus be interpreted as corresponding to the reservoir temperature, which is consistent with chalcedony geothermometry.

## 5. Conclusions

Magumsan thermal waters from six deep wells at temperatures of 30.8–49.8 °C can be chemically classified into two groups: lower-temperature (<31 °C) Na–$HCO_3$ type and higher-temperature (>40 °C) Na–Cl ($HCO_3^-$, $SO_4^{2-}$) type, which are both quite different from local groundwater of Ca–$HCO_3$ type. It is inferred that major ions, such as $Na^+$, $Cl^-$, and $SO_4^{2-}$, and $SiO_2$ in thermal waters are derived from interactions of water with andesitic host rock, pyrite, and chert associated with epithermal activity, and mixing with paleo-seawater in the deep aquifer, which may be clarified by age dating, such as $^{36}Cl$ isotope analysis as further research.

$\delta^{18}O$ and $\delta D$ values of the thermal waters are clustered above the LMWL, resulting in silicate oxidation and $H_2S$ exchange. Isotopic depletion with increasing temperature suggests that isotopically lighter meteoric water that was recharged at higher altitudes migrates into a deeper and higher-temperature thermal reservoir. The $\delta^{34}S$ values of dissolved sulfate in thermal waters can be classified into two groups: low-$\delta^{34}S$ (7.0%–7.2%) and high-$\delta^{34}S$ (9.9%–15%) waters, with the former corresponding to the lower-temperature Na–$HCO_3$ type water and the latter to the higher-temperature Na–Cl ($HCO_3$, $SO_4$) type. The sulfate in higher-$\delta^{34}S$ (>9.9%) waters has two possible sources: kinetic-controlled oxidation of magmatic pyrite in the thermal aquifer causing enrichment in $^{34}S$, and paleo-seawater in the deep aquifer mixing with thermal water of meteoric origin.

In a ternary end-member mixing diagram, the $^3He/^4He$ ratios in thermal waters plot along a single air-mixing line between mantle and crustal origins with a dominant contribution (80.6%–95.7%). A positive correlation between $^4He$ contents and $^4He/^{20}Ne$ ratios and temperatures of thermal water, together with the predominant crustal origin of He, strongly implies that the heat source of thermal waters in the Magumsan area is thermal energy generated by the decay of radioactive elements in crustal rocks.

Magumsan thermal waters plot on the boundary between partial equilibrium and immaturity in a Na–K–Mg ternary diagram, and cation geothermometers and the silica–enthalpy model overestimate the reservoir temperature. A reliable temperature estimate of 55–83 °C was obtained with the thermodynamic equilibrium method, which was consistent with a range of 54–86 °C estimated by chalcedony geothermometry. The mixing ratio of cold shallow water with thermal water is estimated to be 54%–74%.

**Author Contributions:** C.-H.J., B.-D.L., S.-W.A., J.-H.Y., Y.-C.L., Y.-J.L., and H.-W.J. conceived and designed the research project. C.-H.J., K.N., S.-W.A., and K.-H.K. performed the fieldwork and collected the water samples. K.N. and K.-H.K. conducted noble-gas analyses. All authors read and approved the manuscript.

**Funding:** This research was financially supported by a Basic Research Project (19-3411) of the Korea Institute of Geoscience and Mineral Resources (KIGAM), funded by the Ministry of Science and ICT of Korea, and by the Daejeon University research fund (20173610).

**Conflicts of Interest:** The authors declare no conflict of interest.

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
