# Peer review of "Geochemical and Isotopic Compositions and Geothermometry of Thermal Waters in the Magumsan Area, South Korea"

_water, doi:10.3390/w11091774_

Round 1

Reviewer 1 Report

The manuscript concerns the application of geochemical and isotopic methods to explain the origin and formation of thermal waters in Magumsan area of south Korea.

The manuscript is interesting, well written and organized in clear and understandable manner. However, I found some elements which could be improved. My remarks are following:

Introduction: There is lack of information about current stage of knowledge in relation to geothermal waters at Magumsan, what kind of research have been done up to date, and what issues needs to be explore to evaluate the geothermal reservoir. I recommend to add a few words and references about investigations done up to date.

Table 1: ORP values - it is not clear what this values mean? Is this redox potential measured by some electrode? - If yes - the potential of the electrode should be provided for the readers in order to give them possibility to calculate real Eh values (redox normalized to standard hydrogen electrode) to know how is the environment in the aquifer: reduced or oxidized. In actual form it is not know.

Text lines: 201-203: I recommend to reformulate this part and directly refer to two different hydrochemical group of waters - as this group of water can be deduced from Piper diagram. So, avoid to write about some groups of high or low concentrations of ions.

Text lines: 247-350: the same remark as above: reformulate to say directly about two different hydrochemical groups of water but not about higher-level or lower-level groups.

Fig 6: Use "Piper diagram" instead of "trilinear plot"

Fig. 7: This figure it is not a diagram but relationship beewtn d18O and d2H - cgange accordingly.

Table 4: Use "outflow" instead of "outlet". Is there known temperatures measured at the bottom of the thermal wells? If yes - they should be also added to this comparison.

The explanation to Table 4 - placed under the Table: I recommend to provide each mathematical equation of the geothermometers which were used to calculate the values of temperature. Otherwise, it is very hard to check the results or to use this paper to comparison with other calculations from other authors; some geothermometers have a few updates or corrections. In my opinion this work will be better if such equations will be provided by the authors.

Conclusions: I recommend to add what needs to be done in the future to better recognized the geothermal reservoir and formation of thermal water.

Reviewer 2 Report

The study presents some important isotopic and geochemical data of the spring waters from Korean Peninsula to understand the chemical composition, recharge origin, sources of Sulphur and helium and heat sources. However, I also observed five similar (refs 6-9, 21 in the present manuscript) works, mostly by these authors, observed similar results and interpretations (whatever I understood from abstract and figures as most of the texts are in a different language). Can the authors highlight what new they observed in the present study which they did not report in the previous studies? Unfortunately, I cannot recommend this manuscript in its present condition. However, I would like to encourage the authors to resubmit it with detailed comparison with the previous findings and highlighting the new results that were not reported previously.

Specific comments:

Introduction needs to be rearranged. Authors discussed the details of the sites in the introduction (L#36-47). The descriptions of the sites and their geochemical and other information can be given in a separate section. Introduction should highlight the importance of the work, information about previous works in the field, identify the knowledge gap, justification of the use of the proxies to understand the processes being studied here and define the objective(s) of the present study, some of which are missing in the present manuscript. Particularly, these authors have similar studies (e.g., refs. 6-9) in some other sites, they must briefly discuss the previous findings and highlight what extra/new they are reporting in the present study.

L#201-203: Incomplete sentence

L#205-208: define what are NaHCO3, NaCl and Ca-HCO3 type waters

L#226-229, Fig 7: Can you explain how recharge of meteoric water cause enrichment of hydrogen isotopic composition of the thermal water? How exchange of water with silicate or with H2S cause deviation from meteoric water line?

L#235: what is oxygen shift? Do you mean shift in d18O?

L#235-237: Can you explain why no shift in oxygen isotopic composition due to water-rock interactions are observed in the present case and observed in the other studies?

L239-241: what is the justification behind the statement that the depleted isotopic values for higher temperature waters are due to altitude effect? I would expect other way, higher altitude should have lower temperature and depleted isotopic values due to altitude effect. This could also be due to enhanced exchange with silicate or H2S (as claimed by the authors in l#226-229) at higher temperatures.

L#228-229: How silicate hydration or H2S exchange fractionate isotopes in water?

Round 2

Reviewer 1 Report

The authors improved the manuscript according to my suggestions. I am satisfy with the provided improvements and I do not have additional remarks.

I recommend paper to publication in Water.

Reviewer 2 Report

The authors convincingly addressed all the concerns raised on the initial draft. I recommend to publish the revised manuscript in its present form.